# Acceptance of Nirsevimab for the Prevention of Respiratory Syncytial Virus Infection in Neonates: A Cross-Sectional Survey in Emilia-Romagna, Italy

**DOI:** 10.3390/vaccines13090896

**Published:** 2025-08-23

**Authors:** Susanna Esposito, Valentina Fainardi, Maria Elena Capra, Melodie Aricò, Angela Lanzoni, Beatrice Rita Campana, Marta Niceforo, Cosimo Neglia, Enrico Valletta, Giacomo Biasucci, Serafina Perrone

**Affiliations:** 1Pediatric Clinic, University Hospital of Parma, 43126 Parma, Italy; valentina.fainardi@unipr.it (V.F.); beatricerita.campana@unipr.it (B.R.C.); marta.niceforo@studenti.unipr.it (M.N.); negliamino@gmail.com (C.N.); 2Pediatrics and Neonatology Unit, Guglielmo da Saliceto Hospital, 29100 Piacenza, Italy; m.capra@ausl.pc.it (M.E.C.); giacomo.biasucci@unipr.it (G.B.); 3Pediatric Unit, G.B. Morgagni-L. Pierantoni Hospital, AUSL Romagna, 47100 Forlì, Italy; melodieolivialoredanarosa.arico@auslromagna.it (M.A.); enrico.valletta@auslromagna.it (E.V.); 4Pediatric Unit, Santa Maria della Scaletta Hospital, 40026 Imola, Italy; angela.lanzoni@ausl.imola.bo.it; 5Neonatology, University Hospital of Parma, 43126 Parma, Italy; serafina.perrone@unipr.it

**Keywords:** respiratory syncytial virus (RSV), nirsevimab, monoclonal antibodies, parental acceptance

## Abstract

*Background*: Respiratory syncytial virus (RSV) bronchiolitis remains a leading cause of hospitalization in infants, particularly those with risk factors such as prematurity or chronic diseases. Nirsevimab, a long-acting monoclonal antibody, has recently been approved for RSV prevention. However, parental acceptance of this novel immunoprophylaxis is crucial for effective implementation. The aim of this study was to investigate parental acceptance of nirsevimab prophylaxis for RSV among eligible neonates in Emilia-Romagna, Italy, and to identify factors influencing decision making. *Methods*: A prospective, multicenter observational study enrolled 1042 parents of neonates eligible for nirsevimab prophylaxis according to regional criteria. Parents completed a semi-structured questionnaire during pre-immunization counseling, exploring knowledge, attitudes, perceived risks, information sources, and willingness to accept prophylaxis. Statistical analysis assessed associations between parental characteristics and acceptance rates. *Results*: Among the 1042 respondents, 87.0% (n = 907) expressed willingness to administer nirsevimab to their child, while 2.2% (n = 23) refused and 8.8% (n = 92) were undecided. Higher acceptance was significantly associated with awareness of RSV risks (72.1% vs. 41.7%, *p* < 0.01), belief in nirsevimab’s high efficacy (46.2% vs. 18.3%, *p* < 0.01), and lower concern over side effects (10.6% vs. 27.8%, *p* < 0.01). Trust in primary care pediatricians and the healthcare system was also notably higher among accepting parents (*p* < 0.001). Willingness to pay declined with a hypothetical EUR 250 cost but remained higher among the acceptance group (71.0% vs. 50.4%, *p* < 0.001). *Conclusions*: Parental acceptance of nirsevimab in Emilia-Romagna was high, though significant gaps in knowledge and concerns about safety persist. Targeted educational strategies that clarify the nature, efficacy, and safety of nirsevimab—alongside maintaining cost-free access—are essential to support the successful implementation of RSV prophylaxis programs.

## 1. Introduction

Respiratory syncytial virus (RSV) is a ubiquitous pathogen that represents one of the most significant causes of lower respiratory tract infections in young children worldwide, particularly in infants under twelve months of age [1,2,3]. RSV is the primary etiological agent of bronchiolitis, a condition that accounts for a substantial proportion of pediatric hospitalizations, especially during the winter months in temperate climates [4]. Nearly 90% of children contract RSV by the age of two, and reinfections are common due to the incomplete and short-lived immunity elicited by natural infection [5]. In addition to its acute impact, RSV has been linked to significant long-term respiratory morbidity, including an increased risk of recurrent wheezing and the development of asthma in childhood [6].

RSV exhibits pronounced seasonality, with peak incidence in the Northern Hemisphere typically observed between November and April [7]. However, the COVID-19 pandemic profoundly altered the epidemiological patterns of RSV, initially causing a drastic decline in cases due to stringent public health measures such as social distancing and mask mandates [8,9]. Following the relaxation of these measures, many regions experienced atypically intense and earlier RSV seasons, suggesting an “immunity debt” effect whereby reduced viral circulation left a larger susceptible population [10].

Until recently, the only pharmacological prophylaxis available against RSV was palivizumab, a monoclonal antibody requiring monthly intramuscular injections throughout the RSV season [11]. Its use was reserved for infants at high risk of severe disease, including those born preterm and those with significant congenital heart disease, chronic lung disease, or severe immunodeficiencies. However, the high cost and narrow indications of palivizumab limited its widespread application.

In October 2022, the European Medicines Agency approved nirsevimab, a novel monoclonal antibody targeting the RSV fusion (F) protein [12]. Unlike palivizumab, nirsevimab is characterized by an extended half-life, offering season-long protection after a single intramuscular dose. Real-word data showed that nirsevimab significantly reduces hospitalizations due to RSV-related lower respiratory tract infections, not only among high-risk groups but also in healthy term infants [13,14,15,16,17].

For the 2024–2025 RSV season, many Italian regions, including Emilia-Romagna, have adopted new prophylactic strategies integrating nirsevimab into universal or expanded immunization programs. In Emilia-Romagna, nirsevimab has been offered in the neonatology units to all neonates born during the epidemic season, spanning from October 2024 to February 2025, as well as to infants and children identified as being at increased risk for severe RSV disease based on well-defined clinical criteria [18]. These include preterm infants born between March and September 2024 with gestational ages greater than 29 but less than 35 weeks; neonates born at 29 weeks of gestation or less who are under 12 months old at the beginning of the RSV season; children up to 24 months of age with bronchopulmonary dysplasia requiring medical treatment in the preceding 6 months; children with hemodynamically significant congenital heart disease; infants with severe congenital anomalies including neuromuscular or tracheobronchial malformations; children who have undergone cardiac transplantation and are in their first or second RSV season; those with primary or secondary immunodeficiencies; children diagnosed with cystic fibrosis; and infants with a history of post-infectious bronchiolitis obliterans secondary to severe viral respiratory infections during the previous season.

Despite the promising clinical effectiveness of nirsevimab, the success of such prophylactic campaigns hinges upon parental acceptance. Understanding parents’ knowledge, attitudes, and concerns regarding RSV and its preventive strategies is essential, as these factors profoundly influence vaccination uptake and program effectiveness. This study was designed to investigate parental acceptance of nirsevimab in neonates born in Emilia-Romagna, Italy, during the 2024–2025 RSV season and to identify key factors that shape parental decision-making regarding its administration to their children.

## 2. Methods

### 2.1. Study Design and Population

This study was designed as a prospective, multicenter, observational investigation conducted within neonatology units across the Emilia-Romagna region in Italy. Participating centers were located in Parma, Piacenza, Imola, and Forlì. The primary aim was to assess parents’ knowledge, perceptions, and acceptance of nirsevimab prophylaxis for RSV among eligible neonates.

The study population comprised parents or legal guardians of neonates and infants residing in the participating centers who were eligible for nirsevimab during the 2024–2025 RSV season, according to Emilia-Romagna recommendation [18]. It included neonates born between 1 October 2024, and 28 February 2025, regardless of individual risk factors, as well as infants and children with specific conditions that placed them at elevated risk for severe RSV disease. The study was approved by the Ethic Committee of the participating centers; participation required that parents or legal guardians provide written informed consent after being fully informed about the study’s objectives and procedures.

### 2.2. Questionnaire

Data collection was accomplished through the administration of a semi-structured questionnaire developed specifically for this study. The development of the questionnaire involved several methodological steps to ensure its validity, relevance, and comprehensibility. Initially, a systematic review of the existing literature was conducted to gather insights into parental knowledge and perceptions about RSV and prior immunization campaigns, as well as factors influencing acceptance of new prophylactic interventions. Consultations were held with experts in neonatology, pediatrics, epidemiology, and psychology to refine the content and ensure alignment with the study’s objectives.

The questionnaire was organized into thematic sections to facilitate systematic data collection. These sections addressed sociodemographic characteristics; willingness to accept nirsevimab prophylaxis for their child; parental awareness and knowledge of RSV and its potential complications; familiarity with and perceptions of nirsevimab’s efficacy and safety; and trust in healthcare providers. To enhance accessibility and ensure comprehension across diverse populations, the language was simplified, and technical terms were minimized. Recognizing the multicultural composition of Emilia-Romagna, the questionnaire was prepared not only in Italian but also translated into English and Arabic. To ensure linguistic accuracy, a rigorous back-translation process was employed for all non-Italian versions.

The questionnaire was primarily composed of multiple-choice items, with most questions allowing respondents to select one or more options as applicable. Participants were informed that they could leave any question unanswered if they wished. For questions that included an “Other” category, free-text fields were provided to enable respondents to specify additional explanations or alternative answers in their own words.

Prior to full-scale implementation, the questionnaire underwent pilot testing with a sample of thirty parents who were representative of the study population. This group included parents of varying educational levels, linguistic backgrounds, and differing health literacy, as well as parents of both high-risk and low-risk neonates. Feedback from this pilot phase was systematically collected to assess the clarity, relevance, and completeness of the questionnaire items. Based on this feedback, refinements were made to enhance clarity and ensure cultural appropriateness.

Parents were invited to complete the questionnaire during routine pre-immunization counseling sessions held in participating neonatology units.

### 2.3. Statistical Analyses

Statistical analyses were planned to include descriptive, inferential, and multivariate approaches. Descriptive statistics summarized demographic data and key questionnaire responses, presenting continuous variables as means and standard deviations or medians and interquartile ranges, depending on data distribution, and categorical variables as frequencies and percentages. Inferential analyses compared characteristics between parents who accepted or declined nirsevimab prophylaxis, employing Student’s *t*-tests or Mann–Whitney U tests for continuous variables, and chi-square or Fisher’s exact tests for categorical data. A significance threshold of *p* < 0.05 was predefined.

## 3. Results

### 3.1. Study Population

A total of 1042 out of 1410 (73.9%) parents or legal guardians of neonates born during the study period participated in the survey. A total of 368 (26.1%) approached parents or legal guardians declined to complete the questionnaire, most commonly citing a lack of time or interest. Table 1 shows the self-reported sociodemographic characteristics of respondents.

The majority of respondents were mothers (66.79%), with fathers accounting for 31.86%, and both parents responding together in 0.96% of cases. Most parents were between 21 and 40 years old (88.49%). Regarding family size, 51.82% were first-time parents, while 36.47% had two children, and 11.71% had three or more children. Approximately 16.8% of infants were identified as having risk factors for severe RSV disease, the most common being prematurity (14.01%) and congenital heart disease (1.43%). The “other” category for neonatal comorbidities (n = 14; 1.34%) included conditions such as cystic fibrosis (n = 1), severe immunodeficiencies (n = 2), neuromuscular disorders (n = 2), and major congenital anomalies (n = 9) not otherwise specified in the predefined categories.

### 3.2. Knowledge and Perceptions About RSV and Nirsevimab

Table 2 summarizes knowledge and perception about RSV and nirsevimab.

Among all respondents, 68.23% reported awareness of the risks associated with RSV infection. However, only one-third (33.01%) had heard of nirsevimab before being approached during this survey.

When asked what they believed nirsevimab to be, 65.93% correctly identified it as a monoclonal antibody that helps prevent infections, while 23.32% thought it was a vaccine.

Table 3 describes willingness to administer nirsevimab to their child, familiarity with nirsevimab’s efficacy and safety, and trust in healthcare.

Overall, 87.04% of respondents declared their willingness to administer nirsevimab to their child, whereas 2.21% explicitly refused, and 8.83% were undecided. The recommendation of their pediatrician (71.2%), the availability of safety data (43.76%), and the availability of efficacy data (39.26%) were identified as the main factors influencing parents’ decision to administer nirsevimab to their child.

The majority of respondents (83.40%) stated that they would like to receive more information about nirsevimab and its indications, with discussions with their primary care pediatrician (49.71%) being the preferred channel for obtaining such information.

Most respondents reported trusting the recommendations of the national healthcare system regarding nirsevimab (56.72% “very much” and 38.0% “moderately”) and expressed willingness to purchase the medication out-of-pocket at an approximate cost of EUR 250.

### 3.3. Profile of Parents Favorable to Nirsevimab

An analysis was therefore conducted on the sample, stratified between those who were not in favor or were unsure about having nirsevimab administered to their child, and those who were in favor. Table 4 shows the variables that displayed statistically significant differences.

Based on the analyses, parents favorable to nirsevimab were more often aged between 31 and 40, with a university degree, and typically first-time or second-time parents. They were significantly more likely to have prior knowledge about RSV risks and correctly identify nirsevimab as a monoclonal antibody. They perceived their child as moderately to highly at risk of RSV, expressed greater confidence in the efficacy of nirsevimab, and had fewer concerns about its potential side effects. Trust in primary care pediatricians and the healthcare system emerged as critical determinants of their acceptance. Finally, they demonstrated higher willingness to pay for the prophylaxis if necessary, signaling a stronger perceived value of the intervention. Ethnicity distribution was comparable between parents who were not in favor or were unsure about having nirsevimab administered to their child and those who were in favor, with no statistically significant differences observed.

## 4. Discussion

This study offers the first comprehensive insights into parental attitudes toward nirsevimab prophylaxis in Italy, conducted within the context of the Emilia-Romagna region’s upcoming RSV immunization program for in-season-born neonates. The findings reveal a generally high level of acceptance (87.04%) among parents of neonates and infants, suggesting a favorable initial landscape for the roll-out of nirsevimab.

Similar results were observed in other countries [19,20,21], although a closer examination of the data uncovers nuances and challenges that require attention to ensure successful program implementation. For example, a recent multicenter study in France reported that 85% of parents accepted nirsevimab administration in the maternity setting, with safety perceptions and healthcare provider recommendations being key determinants, similar to our findings [21]. In Spain, population-based data from Galicia during the first universal nirsevimab season showed coverage above 90% for eligible infants, reflecting high public trust in regional health authorities and early, proactive communication campaigns [13]. Italian data from Valle d’Aosta [14] and Tuscany [15] also documented strong parental uptake, though most available reports are based on real-world administration rates rather than pre-administration intention surveys. Compared with these experiences, our acceptance rate falls within the upper range but is slightly lower than post-implementation coverage figures, possibly reflecting the “plan–do” gap whereby stated willingness may not always translate into actual uptake. Notably, some Canadian [19] and US studies [20] reported that parental willingness to accept RSV preventive measures, including monoclonal antibodies, was lower (often between 60 and 75%), with hesitancy more strongly influenced by uncertainty about safety and the novelty of the intervention. These differences may be partly attributable to structural factors such as the presence or absence of universal public funding, variability in healthcare provider endorsement, and the degree of prior public awareness about RSV. Our data therefore align with the broader European trend of relatively high acceptance, likely supported by established trust in pediatric care networks and the Italian tradition of integrating new pediatric preventive measures into regional health programs.

Even if overall acceptance was high, only about one-third of parents had heard of nirsevimab prior to the study. Even among those who accepted its use, nearly a third expressed uncertainty regarding its efficacy. This finding reflects an important paradox: willingness to adopt a new intervention is widespread, but it coexists with significant gaps in specific knowledge about what nirsevimab is and how it functions. Nearly 23% of parents mistakenly identified nirsevimab as a vaccine rather than a monoclonal antibody, while a non-negligible portion simply responded “I don’t know” when asked about its mechanism or efficacy.

This uncertainty is not surprising, given the novelty of nirsevimab and the complexity of explaining monoclonal antibody prophylaxis to the public. Similar challenges have been reported in earlier studies on palivizumab, where parents often conflated antibody therapy with vaccination or remained unaware of its preventive purpose [22]. The same confusion could potentially undermine trust or foster suspicion, particularly if misinformation circulates online or via social networks.

One of the most striking findings is the dominant role of primary care pediatricians in shaping parental decisions. Parents willing to administer nirsevimab were significantly more likely to report high trust in their pediatricians and in the healthcare system at large. This aligns with extensive vaccine hesitancy research, which consistently identifies provider recommendations as the most influential factor in vaccine uptake across various contexts [23,24,25]. Parents in the “acceptance” group were also more likely to cite their pediatrician’s advice as the primary reason for choosing nirsevimab, underlining the critical gatekeeping role pediatricians play.

In our survey, nearly one in five respondents (18.7%) indicated that they did not have a pediatrician. This was not due to lack of access to pediatric care in general, but rather to the timing of the survey: these respondents were parents of neonates who had not yet been formally assigned a primary care pediatrician, as assignment typically occurs shortly after discharge from the birth hospital. Nevertheless, the finding underscores the importance of ensuring that early counseling about RSV prevention is not delayed until the first pediatric visit. In Emilia-Romagna, pediatric consultations are universally accessible and free of charge, but the timing of first contact may vary, potentially influencing early decision-making about interventions such as nirsevimab. Strengthening communication pathways between hospital-based neonatology staff and community pediatricians—and ensuring consistent messages are delivered during the immediate postnatal period—may help optimize uptake and reduce disparities related to the timing of provider contact.

Nevertheless, even among hesitant or refusing parents, moderate trust in healthcare providers remained relatively high, suggesting that hesitancy is not necessarily rooted in overt distrust but may instead stem from information gaps, safety concerns, or misperceptions. This subtle distinction is crucial for tailoring communication strategies: rather than combatting outright anti-vaccine sentiment, interventions must focus on clarifying scientific facts and alleviating specific fears.

Concerns regarding potential side effects were a significant differentiator between parents who accepted nirsevimab and those who were hesitant or refused it. Among hesitant parents, 27.83% were “very worried” about side effects compared to just 10.58% among those who accepted the prophylaxis. Interestingly, a large proportion of respondents—even among those favorable to nirsevimab—remained uncertain about what specific side effects to expect, with “fever” and “skin rash” being the most commonly cited concerns. Similar results have been reported in Galicia, underscoring the importance of enhancing educational efforts in future campaigns to address safety concerns, thereby supporting broader RSV immunization coverage in the pediatric population [26].

This pattern reflects a broader phenomenon observed in vaccination research: unfamiliarity with new interventions often breeds generalized anxiety about safety, even in the absence of concrete adverse event reports [27,28]. In the case of nirsevimab, its novelty and the lack of widespread public discourse may exacerbate this uncertainty. Therefore, clear communication about safety data, including real-world experience from other countries, will be vital. Notably, early data from these regions have shown reassuring safety profiles, but these findings need to be widely disseminated in lay-friendly terms [14,15,16,17,29,30,31].

Parents who accepted nirsevimab were significantly more likely to perceive RSV as a serious threat, citing bronchiolitis and potential hospitalization as major concerns. This reflects effective health messaging over recent years, emphasizing RSV’s role in infant morbidity. Yet, a substantial minority of parents still minimized RSV’s risks or were unsure about the virus’s consequences, highlighting an area for targeted education.

Interestingly, the perception of one’s own child’s risk played a major role: parents who saw their child as moderately or highly at risk were more inclined to accept prophylaxis, as previously reported [20]. This finding suggests that general knowledge about RSV is not sufficient; personalized risk perception is a key motivator for proactive behavior. It is also possible that parental perception of their child’s RSV risk was influenced, at least in part, by the strength of healthcare providers’ recommendations, as clinicians may have emphasized nirsevimab more strongly for infants they considered at higher risk of severe disease, thereby reinforcing parents’ sense of vulnerability. However, this means that communication strategies should therefore focus not only on generic messages about RSV, but also on helping parents understand how the virus might specifically impact their child’s health.

One of the most pragmatic findings from this study concerns cost. While acceptance of nirsevimab was high when assumed to be publicly funded, willingness dropped substantially if the prophylaxis required out-of-pocket payment of approximately EUR 250. Although 67.37% overall expressed willingness to pay, this proportion decreased significantly among those who were otherwise hesitant about nirsevimab. These data underscore the risk that even moderate costs could reduce uptake, especially among socioeconomically disadvantaged groups. This is an important consideration for policymakers. Universal, publicly funded programs may be the only viable path to achieving equitable protection and avoiding socioeconomic disparities in RSV prevention. Otherwise, the introduction of nirsevimab could inadvertently exacerbate health inequities, leaving the most vulnerable populations unprotected.

This study allows for a tentative characterization of parents who are more inclined to accept nirsevimab. These parents are often between 31 and 40 years of age, with a university degree, and they are either first-time or second-time parents. They show higher health literacy, are more informed about RSV risks, correctly identify nirsevimab as an antibody, and perceive their child to be at least moderately at risk. Crucially, they express high trust in pediatricians and the healthcare system, and they are more willing to pay for prophylaxis if required.

Such a profile provides valuable insights for targeted communication. Parents with lower educational attainment, younger age, or less direct experience with severe infant illness may require more intensive outreach to ensure equitable uptake. This could include simplified educational materials, culturally adapted messaging, and proactive engagement by trusted healthcare providers during both prenatal and immediate postnatal care. By recognizing and addressing these differences, public health programs can help ensure the equitable uptake of nirsevimab across all sociodemographic groups, thereby maximizing the population-level benefits of RSV prevention strategies.

The principal strengths of this study include its large sample size and detailed stratification of attitudes and beliefs, allowing for a robust analysis of factors influencing parental decisions. Furthermore, the study’s alignment with real-world policy contexts in Emilia-Romagna makes its findings directly relevant for ongoing public health planning. However, several limitations warrant caution. A main limitation of this study is that it assessed only the stated intent of parents to accept nirsevimab prophylaxis, without verifying actual administration to the infant. Consequently, we could not account for the potential “plan–do” gap, in which expressed willingness in a survey may not translate into real-world behavior. Moreover, the survey was conducted in hospital settings, possibly introducing selection bias, as parents attending hospitals may differ systematically from those in the general community in terms of health awareness or anxiety levels. Social desirability bias may have led some parents to express willingness to accept nirsevimab even if they were privately undecided. Additionally, the cross-sectional nature of the data precludes an assessment of how parental attitudes might change over time, especially once nirsevimab becomes more widely known or if safety signals were to emerge post-implementation. However, this study offers essential guidance for the upcoming implementation of nirsevimab programs in Italy. To sustain the high levels of acceptance already observed, public health authorities should take several key actions. First, they need to prioritize education by clearly explaining that nirsevimab is a monoclonal antibody rather than a vaccine [32], emphasizing its safety and efficacy in straightforward, accessible language. Second, trusted messengers such as pediatricians and neonatologists should be equipped with consistent, easy-to-understand messaging and materials to effectively counsel parents [33]. Third, equity must be addressed by ensuring that the prophylaxis remains free of charge, thereby eliminating economic barriers to access [34]. Finally, it is crucial to monitor parental attitudes over time through ongoing surveillance, especially in response to reports of adverse events or changes in media coverage.

## 5. Conclusions

This large, multicenter observational study conducted in Emilia-Romagna provides critical insights into parental attitudes toward nirsevimab prophylaxis RSV in neonates and infants. The overall acceptance of nirsevimab was high, with 87% of parents expressing willingness to administer it to their child. Acceptance was strongly associated with prior awareness of RSV risks, accurate understanding of nirsevimab as a monoclonal antibody, trust in primary care pediatricians and the healthcare system, and a perception of moderate to high risk of RSV in their child.

However, significant knowledge gaps and misconceptions remain. Many parents had not heard of nirsevimab prior to counseling, and a substantial proportion confused it with a vaccine. Concerns about potential side effects—often poorly specified—contributed to hesitancy. Economic considerations also influenced decision-making; acceptance declined markedly when a hypothetical cost of EUR 250 was introduced, highlighting the importance of ensuring universal and free access to the prophylaxis to avoid inequities.

The study underscores the central role of primary care pediatricians as trusted sources of information. Personalized counseling and accessible educational materials that clarify nirsevimab’s mechanism, safety, and benefits will be essential to sustaining public trust and high uptake. Ongoing monitoring of parental perceptions—especially in response to safety data and media narratives—will be crucial on a global scale. These findings provide actionable guidance for national and regional health authorities. By addressing informational, perceptual, and economic barriers, effective implementation of RSV prevention in early childhood can become a feasible goal.

## Figures and Tables

**Table 1 vaccines-13-00896-t001:** Self-reported sociodemographic characteristics of respondents.

Variable	N = 1042	
**Parent(s) who filled in the questionnaire**	**N**	**%**
Father	332	31.86
Mother	696	66.79
Both	10	0.96
Missing	4	0.38
**Age of the parent**		
≤20 years	10	0.96
>20–≤30 years	261	25.05
>30–≤40 years	661	63.44
>40 years	109	10.46
Missing	1	0.10
**Ethnicity**		
Caucasian	936	89.82
African	46	4.41
Asian	39	3.74
Other	18	1.72
Missing	3	0.28
**Education**		
First-grade school	13	1.25
Secondary school	164	15.74
Diploma	414	39.73
Degree	420	40.31
Missing	31	2.98
**Number of children in the family**		
First child	540	51.82
Second child	380	36.47
≥Third child	122	11.71
**Underlying comorbidities in the neonate**		
None	862	82.73
Yes	175	16.79
Missing	5	0.48
**Type of comorbidity in the neonate**		
Prematurity	146	14.01
Congenital cardiopathy	15	1.43
Other	14	1.34

**Table 2 vaccines-13-00896-t002:** Knowledge and perception about RSV and nirsevimab.

Variable	N = 1042	
**Are you aware of the risks associated with RSV**	n	%
No	315	30.23
Yes	711	68.23
**How much do you think your child is at risk of RSV infection?**		
Not at all	68	6.53
A little	266	25.53
Moderately	506	48.56
Very much	162	15.55
Missing	40	3.84
**What do you think are the main risks associated with RSV?**		
I don’t know	170	16.31
No risk	20	1.92
Bronchiolitis	749	71.88
Pneumonia	200	19.19
Hospitalization for respiratory failure	485	46.55
Hospitalization in intensive care unit	268	25.72
Recurrent wheezing	148	14.20
Asthma	54	5.18
**Have you ever heard of nirsevimab before?**		
Yes	344	33.01
No	689	66.12
Missing	9	0.86
**If yes, where did you get information about nirsevimab?**	**n**	%
Internet/TV/Social	109	31.69
Primary care pediatrician	103	29.94
Friends	64	18.60
Other	55	15.99
Neonatologist	52	15.12
Gynecologist	44	12.79
Prenatal class	11	3.20
Hospital	8	2.33
University	3	0.87
**What do you think nirsevimab is?**		
A regular medication	5	3.36
A vaccine	243	23.32
An antibody that helps prevent infections	687	65.93
Other	43	4.13

RSV, respiratory syncytial virus.

**Table 3 vaccines-13-00896-t003:** Willingness to administer nirsevimab to the child, familiarity with nirsevimab, and trust in healthcare.

	N = 1042	
Variable	n	%
**Are you willing to have nirsevimab administered to your child?**		
No	23	2.21
Yes	907	87.04
I’m not sure	92	8.83
**What factors most influence your decision to have nirsevimab administered to your child?**		
Primary care pediatrician’s recommendation	742	71.21
Opinion of friends	59	5.66
Information from media sources	65	6.24
Safety data	456	43.76
Efficacy data	409	39.25
Other	59	5.66
**How effective do you believe nirsevimab is in protecting your child from RSV?**		
I don’t know	381	36.56
Very much	441	42.32
Moderately	203	19.48
A little	3	0.29
Not at all	2	0.19
Missing	12	1.15
**How concerned are you about the possible side effects of nirsevimab?**		
Very much	111	10.65
Moderately	412	39.54
A little	374	35.89
Not at all	130	12.48
Missing	15	1.44
**What do you think might be the side effects of nirsevimab?**		
I don’t know	354	33.97
Fever	608	58.35
Pain	286	27.45
Skin rash	286	27.45
Other	32	3.07
**How important is it for you to receive detailed information about the side effects and risks associated with nirsevimab?**		
Very important	866	83.11
Morerately important	131	12.57
A little important	29	2.78
Not important	8	0.77
Missing	8	0.77
**How much do you trust your pediatrician when it comes to advice about your child’s health?**		
I don’t have a pediatrician yet	195	18.71
Very much	663	63.63
Moderately	160	15.36
A little	10	0.96
Not at all	2	0.19
Missing	12	1.15
**Would you be interested in receiving more information about nirsevimab and its indications?**		
Yes	869	83.40
No	152	14.59
Missing	21	2.02
**How would you prefer to receive information about nirsevimab?**		
Conversation with a neonatologist	270	25.91
Conversation with primary care pediatrician	518	49.71
Conversation with a trusted doctor	93	8.93
Informational material	272	26.10
Internet/social media	104	9.98
Other	40	3.84
**How much do you trust the national healthcare system regarding the safety and effectiveness of nirsevimab?**		
Very much	591	56.72
Moderately	396	38.00
A little	38	3.65
Not at all	5	0.48
Missing	12	1.15
**If the cost of nirsevimab were the family’s responsibility and the medication cost approximately EUR 250, would you be willing to pay?**		
Yes	702	67.37
No	311	29.85
Missing	29	2.78

RSV, respiratory syncytial virus.

**Table 4 vaccines-13-00896-t004:** Variables that showed significant differences between those who were not in favor or were unsure about having nirsevimab administered to their child and those who were in favor.

Variable	Not in Favor or Unsure of Nirsevimab (N = 115)		In Favor of Nisevimab (n = 907)		*p*-Value
**Age of the parent**					**<0.05**
≤20 years	2	1.74	8	0.88	
>20–≤30 years	37	32.17	219	24.15	
30–≤40 years	58	50.43	590	65.05	
>40 years	18	15.65	89	9.81	
Missing			1	0.11	
**Education**					**<0.05**
First-grade school	4	3.48	9	0.99	
Secondary school	23	20.00	137	15.10	
Diploma	50	43.48	358	39.47	
Degree	35	30.43	379	41.79	
Missing	3	2.61	24	2.65	
**Are you aware of the risks associated with RSV?**					**<0.01**
No	65	56.52	244	26.90	
Yes	48	41.74	654	72.11	
**How much do you think your child is at risk of RSV infection?**					**<0.01**
Not at all	19	16.52	47	5.18	
A little	40	34.78	224	24.70	
Moderately	38	33.04	464	51.16	
Very	11	9.57	148	16.32	
Missing	7	6.09	24	2.65	
**What do you think are the main risks associated with RSV?**					
I don’t know	41	35.65	124	13.67	**<0.01**
No risk	4	3.48	14	1.54	0.13
Bronchiolitis	60	52.17	682	75.19	**<0.01**
Pneumonia	16	13.91	182	20.07	0.12
Hospitalization for respiratory failure	30	26.09	450	49.61	**<0.001**
Hospitalization in intensive care unit	17	14.78	251	27.67	**<0.01**
Recurrent wheezing	10	8.70	137	15.10	0.07
Asthma	4	3.48	49	5.40	0.50
**Have you ever heard about nirsevimab before?**					**<0.05**
Yes	29	25.22	315	34.73	
No	86	74.78	587	64.72	
Missing					
**What do you think nirsevimab is?**					
A regular medication	7	6.09	28	3.09	0.07
A vaccine	34	29.57	204	22.49	**<0.05**
An antibody that helps prevent infections	56	48.70	625	68.91	**<0.01**
Other	10	8.70	33	3.64	**<0.01**
**What factors most influence your decision to have nirsevimab administered to your child?**					
Primary care pediatrician’s recommendation	67	58.26	670	73.87	**<0.001**
Opinion of friends	8	6.96	51	5.62	0.53
Information from media sources	8	6.96	57	6.28	0.81
Safety data	43	37.39	408	44.98	0.09
Efficacy data	32	27.83	373	41.12	**<0.01**
Other	10	8.70	49	5.40	0.17
**How effectively do you believe nirsevimab is in protecting your child from RSV?**					**<0.01**
I don’t know	75	65.22	294	32.41	
Very much	21	18.26	419	46.20	
Moderately	17	14.78	184	20.29	
A little	1	0.87	2	0.22	
Not at all	1	0.87	1	0.11	
Missing			7	0.77	
**How concerned are you about the possible side effects of nirsevimab?**					**<0.01**
Very much	32	27.83	96	10.58	
Moderately	48	41.74	379	41.79	
A little	28	24.35	321	35.39	
Not at all	6	5.22	96	10.58	
Missing	1	0.87	8	0.88	
**What do you think might be the side effects of nirsevimab?**					
I don’t know	61	53.04	284	31.31	**<0.001**
Fever	45	39.13	559	61.63	**<0.001**
Pain	25	21.74	259	28.56	0.11
Skin rash	21	18.26	263	29.00	**<0.05**
Other	9	7.83	23	2.54	**<0.01**
**How important is it for you to receive detailed information about the side effects and risks associated with nirsevimab?**					**<0.01**
Very important	90	78.26	764	84.23	
Moderately important	14	12.17	115	12.68	
A little important	7	6.09	21	2.32	
Not important	4	3.48	4	0.44	
Missing			3	0.33	
**Would you be interested in receiving more information about nirsevimab and its indications?**					**<0.001**
Yes	80	69.57	783	86.33	
No	34	29.57	117	12.90	
Missing	1	0.87	7	0.77	
**How would you prefer to receive information about nirsevimab?**					
Conversation with a neonatologist	32	27.83	235	25.91	0.44
Conversation with primary care pediatrician	51	44.35	463	51.05	0.38
Conversation with a trusted doctor	13	11.30	79	8.71	0.27
Informational material	19	16.52	251	27.67	**<0.05**
Internet/Social media	18	15.65	86	9.48	**<0.05**
Other	6	5.22	34	3.75	0.37
**How much do you trust the national healthcare system regarding safety and effectiveness of nirsevimab?**					**<0.001**
Very much	42	36.52	546	60.20	
Moderately	51	44.35	340	37.49	
A little	18	15.65	20	2.21	
Not at all	3	2.61	1	0.11	
Missing	1	0.87			
**If the cost of nirsevimab were the family’s responsibility and the medication cost approximately 250 euros, would you be willing to pay?**					**<0.001**
Yes	58	50.43	644	71.00	
No	53	46.09	249	27.45	
Missing	4	3.48	14	1.54	

RSV, respiratory syncytial virus.

## Data Availability

All the available data are included in the manuscript.

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
