# Peer review of "Acceptance of Nirsevimab for the Prevention of Respiratory Syncytial Virus Infection in Neonates: A Cross-Sectional Survey in Emilia-Romagna, Italy"

_vaccines, 2025, doi:10.3390/vaccines13090896_

Round 1

Reviewer 1 Report

Comments and Suggestions for Authors

As countries around the world implement RSV immunisation programmes, it is crucial to identify and address barriers to public acceptance and uptake. The authors of this study sought to understand these in the Emilia-Romagna region of Italy. Prior to the study, the authors undertook substantial work, including a systematic literature review and consultations with various experts, to design a comprehensive and accessible questionnaire. Additionally, a pilot study with 30 participants was conducted to gather feedback and ensure relevance. After surveying 1,042 participants, the authors identified a range of socio-demographic, clinical, and other factors associated with parents’ decisions to have their children immunised with nirsevimab. The study also outlined a general profile of parents who were in favour of immunisation during the 2024/25 RSV season in this region of Italy. Key limitations of the study, such as potential selection bias and the impact of social desirability, are acknowledged in the discussion. These findings offer valuable insights for optimising RSV immunisation programmes in Italy and in other countries, and will  likely prove a useful resource for those seeking to look into reasons for sub-optimal RSV maternal vaccination uptake, or uptake of other childhood vaccinations.

However, we believe that several aspects of the paper would benefit from further clarification, particularly regarding study design, statistical analysis, and the results reported.

 Major comments

  1. Study design:
    1. Please clarify the format of the questionnaire. Were questions open-ended or multiple-choice (or both)? Could respondents select more than one answer or leave any questions unanswered? For multiple-choice questions, could respondents provide free-text answers such as their own explanations for what they thought nirsevimab was (are these included in “Other” category - line 263)?
    2. Study population: 73.9% of parents/legal guardians participated in the survey (line 175), meaning that more than 25% of eligible participants did not complete the questionnaire. Could the authors elaborate on why they did not participate (e.g., refused or could not participate, or were not invited to participate)?  
    3. Statistical analysis: chi-square/ Fisher’s tests or Student’s t-tests were used to identify characteristics that were significantly (p<0.05) associated with refusing immunisation. However, logistic regression analysis with adjustment for potential confounders would allow for more accurate and detailed results by providing adjusted odds ratios and confidence intervals, rather than p-values in isolation.

  1. Table 4:
    1. The table selectively reports only those variables that were found to be significantly (p<0.05) associated with parents’ willingness to accept nirsevimab for their children. It would be helpful if all tested variables were reported regardless of the p-values. For instance, if ethnicity was not found to be statistically significant in this study, it is still an important result.
    2. Respondents willing to accept nirsevimab (907) and those unwilling/ undecided (115) add up to 1,022 participants. Why were 20 of 1,042 respondents excluded?

  1. Discussion:
    1. How did the results of this study compare to similar studies from Italy and other countries? Are there any notable differences?
    2. Nirsevimab is intended to prevent severe RSV-associated disease, not all RSV infections.
    3. Regarding the important association found between paediatrician recommendation and acceptance of nirsevimab (line 372), it would be helpful to discuss whether paediatrician consultations are equally accessible to everyone in the region, as nearly 19% of respondents indicated that they “don’t have a paediatrician” (Table 3).
    4. Impact of personalised risk perception among parents is discussed (line 414). Could this perception have been impacted by health care providers’ recommending nirsevimab more strongly for infants they considered at higher risk of severe disease?

Minor comments

  • Line 191 (Table 1): How were ethnic groups defined and assigned (e.g. self-reported)? Was “Other” one of the options that participants could select, or is this category used to include several rarely reported ethnicities?
  • Line 202 (Table 1): The variable “number of sons” seems to refer to the overall number of children in the family. If so, please correct it to “number of children” to avoid only referring to male children.
  • Line 210 (Table 1): Please consider providing definitions for comorbidities such as “Prematurity” and expanding on comorbidities included under the “Other” category.
  • Line 227 (Table 2) The total number of participants is given as 1,042, but the number of responses to some questions differ without any explanation. For example, the first question has 1,026 answers. Can you please provide explanations for these differences?
  • Line 260 (Table 2): A total of 978 answers are reported, but percentages do not add up correctly. Please double-check the numbers.
  • Line 321: Section “Profile of parents favourable to nirsevimab” might fit better into the Discussion section, as it goes beyond neutral reporting of the results.
  • Tables 1-4: Please use a consistent approach when reporting decimal numbers, using the same number of decimal places and dots (instead of commas) for decimal points.

Author Response

As countries around the world implement RSV immunisation programmes, it is crucial to identify and address barriers to public acceptance and uptake. The authors of this study sought to understand these in the Emilia-Romagna region of Italy. Prior to the study, the authors undertook substantial work, including a systematic literature review and consultations with various experts, to design a comprehensive and accessible questionnaire. Additionally, a pilot study with 30 participants was conducted to gather feedback and ensure relevance. After surveying 1,042 participants, the authors identified a range of socio-demographic, clinical, and other factors associated with parents’ decisions to have their children immunised with nirsevimab. The study also outlined a general profile of parents who were in favour of immunisation during the 2024/25 RSV season in this region of Italy. Key limitations of the study, such as potential selection bias and the impact of social desirability, are acknowledged in the discussion. These findings offer valuable insights for optimising RSV immunisation programmes in Italy and in other countries, and will  likely prove a useful resource for those seeking to look into reasons for sub-optimal RSV maternal vaccination uptake, or uptake of other childhood vaccinations.
However, we believe that several aspects of the paper would benefit from further clarification, particularly regarding study design, statistical analysis, and the results reported.
Re: Thank you for your positive evaluation. We improved the manuscript according to your suggestions.

 Major comments
1.    Study design:
1.    Please clarify the format of the questionnaire. Were questions open-ended or multiple-choice (or both)? Could respondents select more than one answer or leave any questions unanswered? For multiple-choice questions, could respondents provide free-text answers such as their own explanations for what they thought nirsevimab was (are these included in “Other” category - line 263)?

Re: Clarified (p. 4).

2.    Study population: 73.9% of parents/legal guardians participated in the survey (line 175), meaning that more than 25% of eligible participants did not complete the questionnaire. Could the authors elaborate on why they did not participate (e.g., refused or could not participate, or were not invited to participate)?  

Re: Clarified (p. 5).

3.    Statistical analysis: chi-square/ Fisher’s tests or Student’s t-tests were used to identify characteristics that were significantly (p<0.05) associated with refusing immunisation. However, logistic regression analysis with adjustment for potential confounders would allow for more accurate and detailed results by providing adjusted odds ratios and confidence intervals, rather than p-values in isolation.
     Re: We thank the reviewer for this valuable suggestion. We agree that logistic regression analysis would provide more accurate and detailed results by allowing for adjustment of potential confounders and calculation of adjusted odds ratios with confidence intervals. However, our primary objective in this study was to describe parental acceptance patterns and their unadjusted associations with demographic and attitudinal factors, rather than to build a predictive model. Given the exploratory nature of this initial investigation and the large number of potential variables relative to the number of refusal events (n = 23), performing multivariable logistic regression could have led to unstable estimates. Nevertheless, we recognize the value of such an approach, and future studies with larger numbers of refusal events will be designed to include adjusted analyses.

2.    Table 4:
1.    The table selectively reports only those variables that were found to be significantly (p<0.05) associated with parents’ willingness to accept nirsevimab for their children. It would be helpful if all tested variables were reported regardless of the p-values. For instance, if ethnicity was not found to be statistically significant in this study, it is still an important result.
Re: We thank the reviewer for this helpful suggestion. We agree that presenting all tested variables would provide a more complete picture of the analyses. However, given the length and complexity of Table 4, including all variables within the main text would significantly reduce its readability. However, we added the results on ethnicity distribution (p. 14).

2.    Respondents willing to accept nirsevimab (907) and those unwilling/ undecided (115) add up to 1,022 participants. Why were 20 of 1,042 respondents excluded?
Re: Overall, 907 answered to be in favour of nirsevimab, 23 against, and 92 unsure. The other 23 did not answer. 

3.    Discussion:
1.    How did the results of this study compare to similar studies from Italy and other countries? Are there any notable differences?
Re: A paragraph on this issue was added (p. 15).

2.    Nirsevimab is intended to prevent severe RSV-associated disease, not all RSV infections.
Re: We thank the reviewer for this comment. We agree that nirsevimab is primarily indicated for the prevention of severe RSV-associated disease, particularly RSV-associated lower respiratory tract infections that may result in hospitalization. However, clinical trial and real-world data have also demonstrated that nirsevimab significantly reduces RSV-associated outpatient and emergency department visits, in addition to preventing severe disease requiring hospitalization.

3.    Regarding the important association found between paediatrician recommendation and acceptance of nirsevimab (line 372), it would be helpful to discuss whether paediatrician consultations are equally accessible to everyone in the region, as nearly 19% of respondents indicated that they “don’t have a paediatrician” (Table 3).
Re: Clarified in the Discussion (p. 16).

4.    Impact of personalised risk perception among parents is discussed (line 414). Could this perception have been impacted by health care providers’ recommending nirsevimab more strongly for infants they considered at higher risk of severe disease?
Re: We clarified this issue in the Discussion (p. 17).    

Minor comments
•    Line 191 (Table 1): How were ethnic groups defined and assigned (e.g. self-reported)? Was “Other” one of the options that participants could select, or is this category used to include several rarely reported ethnicities? Re: Clarified (p. 5).
•    Line 202 (Table 1): The variable “number of sons” seems to refer to the overall number of children in the family. If so, please correct it to “number of children” to avoid only referring to male children. Re: Done (p. 6).
•    Line 210 (Table 1): Please consider providing definitions for comorbidities such as “Prematurity” and expanding on comorbidities included under the “Other” category. Re: Clarified (p. 7).
•    Line 227 (Table 2) The total number of participants is given as 1,042, but the number of responses to some questions differ without any explanation. For example, the first question has 1,026 answers. Can you please provide explanations for these differences? Re: As clarified in the Methods, participants were informed that they could leave any question unanswered if they wished (p. 4).
•    Line 260 (Table 2): A total of 978 answers are reported, but percentages do not add up correctly. Please double-check the numbers. Re: As explained in the Methods, the questionnaire was primarily composed of multiple-choice items, with most questions allowing respondents to select one or more options as applicable (p. 4).
•    Line 321: Section “Profile of parents favourable to nirsevimab” might fit better into the Discussion section, as it goes beyond neutral reporting of the results. Re: This profile has been strengthened in the Discussion (p. 18). 
•    Tables 1-4: Please use a consistent approach when reporting decimal numbers, using the same number of decimal places and dots (instead of commas) for decimal points. Re: Corrected.

Reviewer 2 Report

Comments and Suggestions for Authors

The investigators report an ERC-approved cross-sectional KAP survey of parents’ (n=1,042) reported acceptance of nirsevimab for their nirsevimab-eligible neonate or infant. The survey response rate was 73.9%. They found that 87% of parents stated they would accept nirsevimab for their neonate or infant; 9% were unsure; and 2% would refuse. They found several factors associated with acceptance. Knowledge of RSV was associated with acceptance, as was trust in their pediatrician. They concluded that “parental acceptance of nirsevimab in Emilia-Romagna was high, though significant gaps in knowledge and concerns about safety persist.” They suggest that “targeted educational strategies that clarify the nature, efficacy, and safety of nirsevimab—alongside maintaining cost-free access—are essential to support successful implementation of RSV prophylaxis programs.”

Acceptance of nirsevimab among parents of eligible infants is clearly important, since nirsevimab only works when given. The use of a cross-sectional survey is a reasonable way to determine reported acceptance, although actual acceptance would have been a more informative outcome. The analyses are appropriate; the writing is clear. Their conclusions are supported by the data presented, as are their suggestions.

I have a few suggestions to improve this manuscript.

The title seems a misnomer, since the authors did not report an observational study. Rather, the study was a cross-sectional survey. Had they observed the parents or infants for any outcome, it could have been called an observational study. However, all data were from a survey. The title should be revised to reflect the actual study design.

The main weakness of the study was that the investigators only studied stated intent, not actual practice – whether the infant was given nirsevimab. This should be listed as a study limitation. Determining the actual outcome would have made this a stronger study, since stated intent does not address the “plan-do” gap in which some people don’t do what they say they will do in a survey.

Author Response

The investigators report an ERC-approved cross-sectional KAP survey of parents’ (n=1,042) reported acceptance of nirsevimab for their nirsevimab-eligible neonate or infant. The survey response rate was 73.9%. They found that 87% of parents stated they would accept nirsevimab for their neonate or infant; 9% were unsure; and 2% would refuse. They found several factors associated with acceptance. Knowledge of RSV was associated with acceptance, as was trust in their pediatrician. They concluded that “parental acceptance of nirsevimab in Emilia-Romagna was high, though significant gaps in knowledge and concerns about safety persist.” They suggest that “targeted educational strategies that clarify the nature, efficacy, and safety of nirsevimab—alongside maintaining cost-free access—are essential to support successful implementation of RSV prophylaxis programs.”
Acceptance of nirsevimab among parents of eligible infants is clearly important, since nirsevimab only works when given. The use of a cross-sectional survey is a reasonable way to determine reported acceptance, although actual acceptance would have been a more informative outcome. The analyses are appropriate; the writing is clear. Their conclusions are supported by the data presented, as are their suggestions.
Re: Thank you for your positive evaluation. We revised the manuscript according to your suggestions and those received by the other reviewer.

I have a few suggestions to improve this manuscript.
The title seems a misnomer, since the authors did not report an observational study. Rather, the study was a cross-sectional survey. Had they observed the parents or infants for any outcome, it could have been called an observational study. However, all data were from a survey. The title should be revised to reflect the actual study design.
Re: You are right, we changed the title as suggested.

The main weakness of the study was that the investigators only studied stated intent, not actual practice – whether the infant was given nirsevimab. This should be listed as a study limitation. Determining the actual outcome would have made this a stronger study, since stated intent does not address the “plan-do” gap in which some people don’t do what they say they will do in a survey.
Re: We acknowledged that our study assessed only parents’ stated intention to accept nirsevimab prophylaxis, without verifying whether the prophylaxis was actually administered.

Round 2

Reviewer 1 Report

Comments and Suggestions for Authors

We thank the authors for their thoughtful responses to our comments and for the revisions they have made to the manuscript. The clarifications regarding the study design have improved clarity and made the paper easier to follow. The Discussion has been strengthened by including comparisons with data from similar studies conducted in other countries and by providing additional context on the paediatric healthcare system in the Emilia-Romagna region.

Although some of our suggestions, such as conducting logistic regression analysis and reporting of all tested variables, were not fully incorporated, we understand the arguments you have provided and find them reasonable.

Thomas Williams, Clinical Research Fellow, University of Edinburgh

Daira Trusinska, PhD Candidate, University of Edinburgh